# Controllable Continual Test-Time Adaptation

## Abstract

Continual Test-Time Adaptation (CTTA) is an emerging and challenging task where a model trained in a source domain must adapt to continuously changing conditions during testing, without access to the original source data. CTTA is prone to error accumulation due to uncontrollable domain shifts, leading to blurred decision boundaries between categories. Existing CTTA methods primarily focus on suppressing domain shifts, which proves inadequate during the unsupervised test phase. In contrast, we introduce a novel approach that guides rather than suppresses these shifts. Specifically, we propose **C**ontrollable **Co**ntinual **T**est-**T**ime **A**daptation (C-CoTTA), which explicitly prevents any single category from encroaching on others, thereby mitigating the mutual influence between categories caused by uncontrollable shifts. Moreover, our method reduces the sensitivity of model to domain transformations, thereby minimizing the magnitude of category shifts. Extensive quantitative experiments demonstrate the effectiveness of our method, while qualitative analyses, such as t-SNE plots, confirm the theoretical validity of our approach. Our code is available at https://anonymous.4open.science/r/C-CoTTA-BC4F/.

## 1 Introduction

Continual Test-Time Adaptation (CTTA) (Wang et al., 2022) is becoming an emerging field, which explores the adaptability of any machine learning model during test time in dynamic environments. The primary objective of CTTA is to enable a pretrained model to adapt to continuously changing scenarios, where the distribution of data shifts over time. CTTA is practical in many long-term intelligent applications, such as autopilot (Hu et al., 2022; O'Kelly, 2021; Chen, 2020), monitoring (Geiger et al., 2013), medical image analysis (Chen et al., 2023; Gonzalez et al., 2020), where models need to remain robust and accurate against possible changes over extended periods.

The main challenge of CTTA is the accumulation of errors caused by the lack of real labels and the continuous domain shifts. Existing methods mainly avoid error accumulation by using strategies such as Mean Teacher (Tarvainen & Valpola, 2017), augmentation-averaged predictions (Wang et al., 2022; Lyu et al., 2024; Liu et al., 2023), and selecting reliable samples (Yang et al., 2023; Niloy et al., 2024; Niu et al., 2022; Wang et al., 2024) to minimize the impact of domain shift. However, these methods focus on suppressing the shifts, few methods explicitly attempt to guide or control the shifts. This is because the lack of labels in the target domain and the inability to obtain source domain data in CTTA. We evaluate this on CIFAR10C, as shown in Fig. 1(a), we find that in the domain adaptation process, multiple categories to lean towards a confusion region. In fact, CTTA meets continuous unknown domain shifts, making this shift almost uncontrollable. This phenomenon will worsen with the changing domain over time, resulting in error accumulation.

Therefore, *we hypothesize that instead of attempting to suppress the shift, guiding or controlling the shift to remain class-separable may also achieve effective adaptation*. On top of this, the main focus of this paper is to study how to achieve controllable domain shift in CTTA. Generally, controlling domain shifts means that controlling the direction of the shift in feature space. To achieve this, we first need to accurately represent the direction of domain shift. Inspired by the interpretable machine learning (Kim et al., 2018), we represent shift directions using the tool of Concept Ativation Vectors (CAV) (Pahde et al., 2022), which represents the transformation path from one concepts to another. In CTTA, the CAV can be represented by the vector from one prototype to another. With CAV, for a specific category, we obtain its domain shift direction by subtracting the category prototype of the target domain in the feature space from the category prototype of the source domain.

Figure 1: t-SNE visualization of controllable domain shift in CTTA. (a) For CoTTA (Wang et al., 2022), due to the lack of control over domain shift, categories being biased towards others, resulting in fuzzy classification boundaries. (b) In contrast, our method achieves controllable domain shift, so even if categories are shift, it will not lead to confusion among categories.

To further control domain shift, we construct the Domain Shift Controlling Loss (DSCL) loss and the Class Shift Controlling Loss (DSCL) loss. DSCL refers to controlling the shift of the overall domain by constraining the model's sensitivity in that direction, thus reducing the impact of domain shift on model performance. DSCL controls the shift of specific categories by constraining the shift direction of each category to avoid biasing other categories. As shown in Fig. 1(b), our method achieve controllable domain shift, and the direction of the shift will not blur the classification boundaries. Extensive experiments are conducted on three large-scale benchmark datasets to validate the effectiveness of the proposed C-CoTTA framework in various challenging and realistic scenarios.

Our contributions are three-fold:

(1) We evaluate and find that only suppressing domain shifts is insufficient, which may lead to blurred classification boundaries. In contrast, we propose to guide and control shifts to keep the class-separability.

(2) We propose a simple and effective direction representation based on Concept Activation Vectors (CAV) in interpretable machine learning, which utilizes the difference between two prototypes in the feature space.

(3) We propose to explicitly control the direction of specific category bias by preventing any category from leaning towards other categories, in order to prevent the blurring of classification boundaries; at the same time, by reducing the sensitivity of the model to domain shift, we control the overall domain shift to alleviate the impact of domain shift on domain adaptation.

## 2 RELATED WORK

### 2.1 CONTINUAL TEST-TIME ADAPTATION

Continual Test-Time Adaptation (CTTA) (Wang et al., 2022) is an emerging paradigm within the machine learning community designed to address the dynamic nature of real-world data distributions. Unlike traditional Test-Time Adaptation (TTA) (Jain & Learned-Miller, 2011; Sun et al., 2020; Wang et al., 2020), which typically assumes a fixed target domain in a source-free and online manner, CTTA operates under the assumption that the target domain may change over time. The main challenge in CTTA is the potential for catastrophic forgetting and error accumulation. During the test time, as the model adapts to new distributions, it risks losing previously learned knowledge, which can result in a degradation in performance known as catastrophic forgetting (Van de Ven & Tolias, 2019). Moreover, the utilization of pseudo-labels derived from the model's own predictions can introduce errors, which may accumulate over time (Li & Hoiem, 2017; Wang et al., 2022), especially when there are frequent domain shifts.

To address the challenges of error accumulation in CTTA, researchers have developed various strategies. A number of works (Wang et al., 2022; Lyu et al., 2024; Liu et al., 2023) employ augmentation-averaged predictions for the teacher model to boost the teacher's confidence, while others (Chakrabarty et al., 2023; Döbler et al., 2023) add perturbations to the student to enhance the

model's robustness. Various methods (Yang et al., 2023; Niloy et al., 2024; Niu et al., 2022; Wang et al., 2024) focus on selecting reliable samples to eliminate the impact of misclassified samples on domain adaptation. As to the challenge of catastrophic forgetting, Wang et al. (Wang et al., 2022) and Brahma et al. (Brahma & Rai, 2023) believe that the source model is more reliable, thus they designed to restore the source parameters. While these studies address the CTTA issue at the model level, other research efforts (Gan et al., 2023; Yang et al., 2023; Ni et al., 2023) leverage visual domain prompts or a limited subset of parameters to extract ongoing target domain knowledge. However, these approaches primarily focus on suppressing domain shift and there are few methods that explicitly attempt to guide or control domain shift.

## 2.2 CONCEPT ACTIVATION VECTORS

Concept Activation Vectors (CAVs) is an interpretability tool for explaining decision-making processes in deep learning models. Originally, the authors of Kim et al. (2018) define CAV as the normal to a hyperplane that separates examples without a concept from examples with a concept in the model's latent activations. This hyperplane is commonly computed by solving a classification problem, for example, using Support Vector Machines (SVMs) Anders et al. (2022), ridge (Cortes & Vapnik, 1995), lasso (Pfau et al., 2021) or logistic regression (Yuksekgonul et al., 2022). Given its ability to effectively orient concepts, CAVs have been employed for a plethora of tasks in recent years, such as concept sensitivity testing (Kim et al., 2018), model correction for shortcut removal (Anders et al., 2022; Pahde et al., 2023; Dreyer et al., 2023), knowledge discovery by investigation of internal model states (McGrath et al., 2022), and training of post-hoc concept bottleneck models (Yuksekgonul et al., 2022). However, common regression-based methods tend to deviate from the true conceptual direction due to factors such as noise in the data (Haufe et al., 2014). To that end, signal-pattern-based CAVs (referred to signalCAVs) have been proposed (Pahde et al., 2022), which are more robust against noise (Weber et al., 2023; Dreyer et al., 2024; Samek, 2023; Biecek & Samek, 2024). However, during test-time, we may not have access to all samples of a prototype, and due to the lack of true labels, misclassified samples may contaminate the prototype. Therefore, the construction of a prototype is different from interpretable machine learning.

# 3 METHODOLOGY

## 3.1 PROBLEM DEFINITION

Given a classification model pre-trained on a source domain, CTTA methods adapt the source model to the unlabeled target data, where the domain continuously changes. The unsupervised dataset of target domains are denoted as $\mathcal{D}^k = \{x_m^k\}_{m=1}^{N^k}$, where $k$ is the target domain index. As shown in Figure 1, in the process of CTTA, if the domain shifts are uncontrolled, some categories may generate bias towards other categories, resulting in blurred classification boundaries. In this paper, we propose to explicitly control over the shift direction in CTTA. specific categories and the overall domain. In the following, we first study how to represent domain shifts in Sec. 3.2. Then, we propose to control the shift within the process of CTTA in Sec. 3.3.

## 3.2 REPRESENTING SHIFT VIA CAV

Domain shift refers to the distribution shift of each class that occurs in the feature space, generally caused by differences between the target domain distribution in the testing phase and the source domain distribution in the training phase. Therefore, how to represent the domain shift direction in the feature space in CTTA is a challenge.

In the field of interpretable machine learning (Kim et al., 2018; McGrath et al., 2022), CAV (Kim et al., 2018) refers to the normal to a hyperplane that separates examples without a concept from examples with a concept in the model's latent activations feature space. CAV is widely used in areas such as model correction for shortcut removal (Anders et al., 2022; Pahde et al., 2023; Dreyer et al., 2023). The concept in CAV generally refers to high-level semantic information, such as whether there are a large number of striped structures in an image. For example, in the concept of stripe, the label for features extracted from images with stripes is 1, and without stripes is 0.

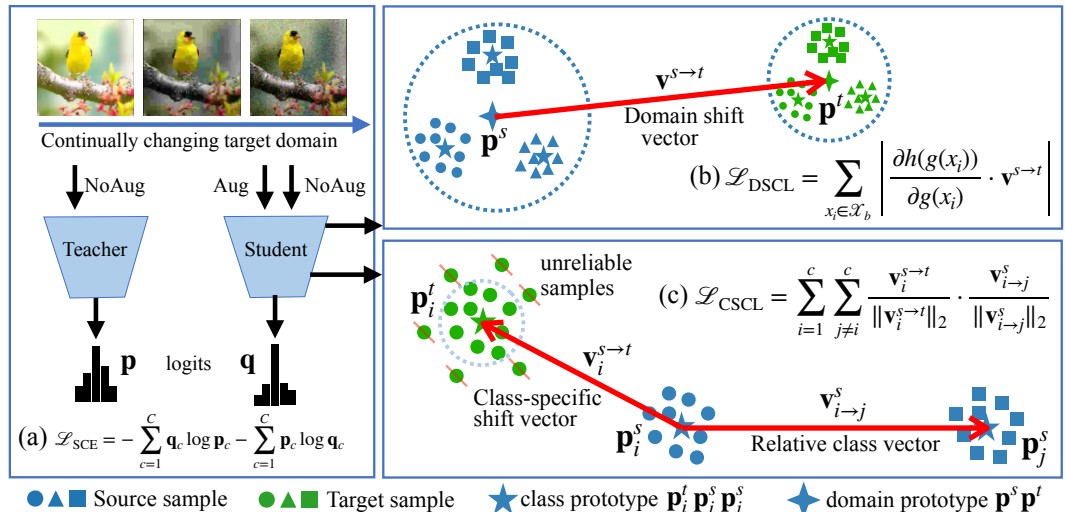

Figure 2: The pipeline of C-CoTTA. (a) Based on the mean teacher framework, perturb the student to enhance model robustness, while optimizing using symmetric cross-entropy. (b) Control the overall domain shift by constraining the model's sensitivity to domain shift directions. (c) Control the shift of a specific category by directly controlling the shift direction of any category to prevent bias towards other categories.

CAV can be calculated in different methods, we use the signal-pattern-based CAVs (SCAV) (Pahde et al., 2022) which provide a simple but effective way to represent the CAV $\mathbf{v}$ as follows:

$$\mathbf{v} = \frac{\text{cov}[f(x), t]}{\text{cov}[t, t]} = \frac{\sum (f(x_i) - \overline{f}(x))(t_i - \overline{t})}{\sum (t_i - \overline{t})^2}, \quad t_i = \begin{cases} 1 & \text{if } x_i \in \mathcal{X}_c, \\ 0 & \text{if } x_i \in \mathcal{X}_n. \end{cases}, \quad (1)$$

where $f(\cdot)$ represents the feature extractor and $t$ represents the concept label of the features, $\mathcal{X}_c$ and $\mathcal{X}_n$ denote sets of samples that either possess or lack the specified concept, respectively. Mean feature $\overline{f}(x) = \frac{1}{N} \sum f(x_i)$ and mean label $\overline{t} = \frac{1}{N} \sum t_i$.

Nevertheless, in the traditional CAV, the concept label $t$ is manually annotated offline. In contrast, in unsupervised CTTA scenario, to represent the domain shift direction in an online manner is intractable. Therefore, we use the pseudo-label $y$ obtained by the existing model to represent concepts automatically. The shift representation for the direction in CTTA can be computed as follows:

$$\mathbf{v} = \frac{\text{cov}[f(x), y]}{\text{cov}[y, y]} = \frac{\sum (f(x_i) - \overline{f}(x))(y_i - \overline{y})}{\sum (y_i - \overline{y})^2}, \quad y_i = \begin{cases} 1 & \text{if } x_i \in \mathcal{X}_t, \\ 0 & \text{if } x_i \in \mathcal{X}_s. \end{cases}, \quad (2)$$

where $\mathcal{X}_t$ and $\mathcal{X}_s$ represent two different data sets (which may express two different categories or two different domains). At the same time, through deduction (refer to Appendix A for details), it is determined that the direction can be further represented as the difference between two prototypes in the feature space.

$$\mathbf{v} = \frac{1}{|\mathcal{X}_t|} \sum_{x_i \in \mathcal{X}_t} f(x_i) - \frac{1}{|\mathcal{X}_s|} \sum_{x_i \in \mathcal{X}_s} f(x_i) = \mathbf{p}^t - \mathbf{p}^s. \quad (3)$$

Based on this, we can calculate various directional vectors, in order to further control domain shift.

**Domain-level source-to-target shift.** We represent the shift of the overall domain as below:

$$\mathbf{v}^{s \to t} = \mathbf{p}^t - \mathbf{p}^s, \quad (4)$$

This calculates by subtracting the target domain prototype $\mathbf{p}^t$ composed of all samples in each batch at test-time from the source domain prototype $\mathbf{p}^s$ composed of prototypes of all categories in the source domain as shown in Figure 2(b). Using the source domain prototype does not violate the CTTA setup, which does not lead to privacy leakage and some previous methods also directly use the source domain prototype, such as RMT Döbler et al. (2023) and SATA Chakrabarty et al. (2023).

**Class-level source-to-target shift.** We also represent the shift of a specific category $i$ as below:

$$\mathbf{v}_i^{s \to t} = \mathbf{p}_i^t - \mathbf{p}_i^s, \tag{5}$$

where we construct it by subtracting the prototype $\mathbf{p}_i^t$ composed of trustworthy samples belonging to that category in each batch at test time from the source domain prototype $\mathbf{p}_i^s$ of that category as shown in Figure 2(c). Moreover, during test time, some samples may be misclassified due to the lack of real labels, leading to distortion in the extracted prototypes. Therefore, we estimate the entropy $H(x^t)$ for each sample $x^t$ from the target domain using the model. We then set aside samples with an entropy exceeding a predefined threshold $E_0$ following Niu et al. (2022).

**Class-to-class source shift.** We then construct class relative shift from category $i$ to $j$ as below:

$$\mathbf{v}_{i \to j}^s = \mathbf{p}_j^s - \mathbf{p}_i^s. \tag{6}$$

As shown in Figure 2(c), this is calculated by using the source domain category prototypes $\mathbf{p}_i^s$ and $\mathbf{p}_j^s$, representing the inherent domain difference between source classes.

## 3.3 Control of shift

We control the shift at domain and class levels. On one hand, the shift of different categories are influenced by the overall domain shift, which is represented by closely aligned feature distributions. This means that after domain shift, the distances between different categories in the feature space become closer. This finding is similar to the conclusions drawn in studies such as Xu et al. (2019); Kondo (2022); Rahman et al. (2021), where researchers found that the feature norms in the target domain are relatively small. On the other hand, the shift of different categories is related to their characteristics and may show different shifts.

**Domain-level shift controlling.** First, we propose to control the shift of the overall domain at the domain level. Specifically, we model the shift sensitivity by considering the gradient of the model output $h(g(x_i))$ with respect to the feature map $g(x_i)$, and then combine it with the overall domain shift direction $\mathbf{v}^{s \to t}$. We can then reduce the sensitivity of the model to domain shift by constraining the direction gradient as follows:

$$\mathcal{L}_{\text{DSCL}} = \sum_{x_i \in \mathcal{X}_b} \left| \frac{\partial h(g(x_i))}{\partial g(x_i)} \cdot \mathbf{v}^{s \to t} \right|, \tag{7}$$

where $h(\cdot)$ represents the remaining part of the model. Intuitively, the Domain Shift Controlling Loss (DSCL) loss $\mathcal{L}_{\text{DSCL}}$ enforces the model output to not change when slightly adding or removing activations along the bias direction as follows:

$$\lim_{\epsilon \to 0} \frac{h(g(\mathbf{x}) + \epsilon \mathbf{v}^{s \to t}) - h(g(\mathbf{x}))}{\epsilon} = 0. \tag{8}$$

Thus, by minimizing $\mathcal{L}_{\text{DSCL}}$, the model becomes insensitive towards the domain shift direction, thereby reducing the impact of domain shift on the domain adaptation process.

**Class-level shift controlling.** Then, in order to prevent uncontrollable shifts of each category, we propose to control class-level shift to avoid any category leaning towards other categories. This requires that the direction of shift $\mathbf{v}_i^{s \to t}$ for any category is preferably the opposite direction of the direction $\mathbf{v}_{i \to j}^s$ of other categories relative to that category as shown in Fig. 2(c). This means that the dot product of $\mathbf{v}_i^{s \to t}$ and $\mathbf{v}_{i \to j}^s$ should be as small as possible. The Class Shift Controlling Loss (CSCL) loss is calculated as follows:

$$\mathcal{L}_{\text{CSCL}} = \sum_{i=1}^c \sum_{j \neq i}^c \frac{\mathbf{v}_i^{s \to t}}{\|\mathbf{v}_i^{s \to t}\|_2} \cdot \frac{\mathbf{v}_{i \to j}^s}{\|\mathbf{v}_{i \to j}^s\|_2}, \tag{9}$$

where we normalize $\mathbf{v}_i^{s \to t}$ and $\mathbf{v}_{i \to j}^s$. The loss $\mathcal{L}_{\text{CSCL}}$ is to prevent any category from shifting towards other categories, achieving controllable domain shift, effectively preventing the decrease in classification performance caused by blurred category boundaries in continuous domain adaptation.

---

**Algorithm 1** Controllable Continual Test-Time Adaptation

---

**Require:** Target domains data $\mathcal{D}^k = \{x_m^k\}_{m=1}^{N^k}$, Source model, Source domain class prototype $\mathbf{p}_i^s$
1: Generate the category relative direction vectors $\mathbf{v}_{i \to j}^s = \mathbf{p}_j^s - \mathbf{p}_i^s$ before domain adaptation
2: **for** a domain $k$ in $K$ **do**
3:     **for** a batch $\{x_b^k\}_{b=1}^B$ in $\mathcal{D}^k$ **do**
4:        Forward the batch, make predictions and get features
5:        Identify reliable samples with low entropy using a predefined threshold $E_0$
6:        Compute the direction of the domain shift $\mathbf{v}^{s \to t} = \mathbf{p}^t - \mathbf{p}^s$ and constrained via Eq. 7
7:        Compute the direction of the class shift $\mathbf{v}_i^{s \to t} = \mathbf{p}_i^t - \mathbf{p}_i^s$ and constrained via Eq. 9
8:        Compute the symmetric cross-entropy loss via Eq. 10
9:        Optimize model by minimizing $\mathcal{L}$ via Eq. 11 and update student and teacher models
10:    **end for**
11: **end for**

---

### 3.4 OVERALL OBJECTIVE

Our work utilizes the symmetric cross-entropy (SCE) loss $\mathcal{L}_{\text{SCE}}$ following Döbler et al. (2023), which is based on the mean teacher framework, involving simply averaging the weights of a student model over time. The resulting teacher model provides a more accurate prediction function than the final function of the student, meanwhile perturbs the student to enhance the robustness of the model (Xie et al., 2020; Sohn et al., 2020). The SCE loss (Wang et al., 2019) which has superior gradient properties compared to the commonly used cross-entropy loss. For enhancing the output of students $q$ and teachers $p$, the SCE loss is defined as follows:

$$\mathcal{L}_{\text{SCE}} = -\sum_{c=1}^C \mathbf{q}_c \log \mathbf{p}_c - \sum_{c=1}^C \mathbf{p}_c \log \mathbf{q}_c. \qquad (10)$$

The overall objective of our proposed continual test-time adaptation method is as follows:

$$\mathcal{L} = \mathcal{L}_{\text{SCE}} + \lambda_1 \mathcal{L}_{\text{DSCL}} + \lambda_2 \mathcal{L}_{\text{CSCL}}, \qquad (11)$$

where $\lambda_1$, and $\lambda_2$ are the hyperparameters. $\mathcal{L}_{\text{DSCL}}$ refers to controlling the shift of the overall domain, which constrains the model's sensitivity in that direction. $\mathcal{L}_{\text{CSCL}}$ controls the shift of specific categories, which constrains the shift direction of each category to avoid biasing other categories. The overall frame diagram is shown in Figure 2.

We illustrate the whole algorithm in Algorithm 1. First, before domain adaptation begins, we use the source domain category prototypes to calculate the inter-class relative direction vector $\mathbf{v}_{i \to j}^s$. During domain adaptation, on one hand, we calculate the shift direction $\mathbf{v}_i^{s \to t}$ for specific categories and constrain it through the loss $\mathcal{L}_{\text{CSCL}}$; on the other hand, we calculate the shift direction $\mathbf{v}^{s \to t}$ for the entire domain and constrain it through the loss $\mathcal{L}_{\text{DSCL}}$. Additionally, we compute the symmetric cross-entropy loss for the prediction logits of the student and teacher, optimize it via Eq. 11, and update the student and teacher models.

## 4 EXPERIMENTS

### 4.1 EXPERIMENTAL SETTING

**Datasets.** We evaluate our proposed method on three CTTA benchmark datasets, including CIFAR10-C, CIFAR100-C, and ImageNet-C. Each dataset contains 15 types of corruptions with 5 levels of severity, ranging from 1 to 5. For simplicity in tables, we use Gauss., Impul., Defoc., Brit., Contr., Elas. and Pix. to represent Gaussian, Impulse, Defocus, Brightness, Contrast, Elastic,and Pixelate, respectively.

**Pretrained Model.** Following previous studies (Wang et al., 2020; 2022), we adopt the pretrained WideResNet-28 (Zagoruyko & Komodakis, 2016), ResNeXt-29 (Xie et al., 2017) and ResNet-50 (He et al., 2016) for CIFAR10-C, CIFAR100-C and Imagenet-C, respectively. Similar to CoTTA, we update all the trainable parameters in all experiments.

**Methods to be Compared.** We compare our C-CoTTA with the original model (Source) and multiple state-of-the-art (SOTA) methods such as BN (Li & Hoiem, 2017; Schneider et al., 2020), TENT (Wang

Table 1: Classification error rate (%) for standard CIFAR10-C continual test-time adaptation task.

| Method | Gauss. | Shot | Impul. | Defoc. | Glass | Motion | Zoom | Snow | Frost | Fog | Brit. | Contr. | Elas. | Pix. | Jpeg | Mean |
|---|---|---|---|---|---|---|---|---|---|---|---|---|---|---|---|---|
| Source | 72.3 | 65.7 | 72.9 | 46.9 | 54.3 | 34.8 | 42.0 | 25.1 | 41.3 | 26.0 | 9.3 | 46.7 | 26.6 | 58.5 | 30.3 | 43.5 |
| BN Li & Hoiem (2017) | 28.1 | 26.1 | 36.3 | 12.8 | 35.3 | 14.2 | 12.1 | 17.3 | 17.4 | 15.3 | 8.4 | 12.6 | 23.8 | 19.7 | 27.3 | 20.4 |
| TENT Wang et al. (2020) | 24.8 | 20.6 | 28.5 | 15.1 | 31.7 | 17.0 | 15.6 | 18.3 | 18.3 | 18.1 | 11.0 | 16.8 | 23.9 | 18.6 | 23.9 | 20.1 |
| CoTTA Wang et al. (2022) | 24.5 | 21.5 | 25.9 | 12.0 | 27.7 | 12.2 | 10.7 | 15.0 | 14.1 | 12.7 | 7.6 | 11.0 | 18.5 | 13.6 | 17.7 | 16.3 |
| RoTTA Yuan et al. (2023) | 30.3 | 25.4 | 34.6 | 18.3 | 34.0 | 14.7 | 11.0 | 16.4 | 14.6 | 14.0 | 8.0 | 12.4 | 20.3 | 16.8 | 19.4 | 19.3 |
| RMT Döbler et al. (2023) | 24.1 | 20.2 | 25.7 | 13.2 | 25.5 | 14.7 | 12.8 | 16.2 | 15.4 | 14.6 | 10.8 | 14.0 | 18.0 | 14.1 | 16.6 | 17.0 |
| PETAL Brahma & Rai (2023) | 23.7 | 21.4 | 26.3 | 11.8 | 28.8 | 12.4 | 10.4 | 14.8 | 13.9 | 12.6 | **7.4** | 10.6 | 18.3 | 13.1 | 17.1 | 16.2 |
| SATA Chakrabarty et al. (2023) | 23.9 | 20.1 | 28.0 | 11.6 | 27.4 | 12.6 | 10.2 | 14.1 | 13.2 | 12.2 | **7.4** | 10.3 | 19.1 | 13.3 | 18.5 | 16.1 |
| DSS Wang et al. (2024) | 24.1 | 21.3 | 25.4 | 11.7 | 26.9 | 12.2 | 10.5 | 14.5 | 14.1 | 12.5 | 7.8 | 10.8 | 18.0 | 13.1 | 17.3 | 16.0 |
| SWA Yang et al. (2023) | 23.9 | 20.5 | 24.5 | **11.2** | 26.3 | **11.8** | 10.1 | 14.0 | 12.7 | **11.5** | 7.6 | **9.5** | 17.6 | 12.0 | 15.8 | 15.3 |
| Ours | **22.7** | **17.9** | **23.8** | 11.6 | 24.3 | 12.8 | **9.5** | 13.1 | 12.4 | 11.6 | 8.0 | **9.5** | 16.4 | 11.4 | 15.6 | **14.7** |

Table 2: Classification error rate (%) for standard CIFAR100-C continual test-time adaptation task.

| Method | Gauss. | Shot | Impul. | Defoc. | Glass | Motion | Zoom | Snow | Frost | Fog | Brit. | Contr. | Elas. | Pix. | Jpeg | Mean |
|---|---|---|---|---|---|---|---|---|---|---|---|---|---|---|---|---|
| Source | 73.0 | 68.0 | 39.4 | 29.3 | 54.1 | 30.8 | 28.8 | 39.5 | 45.8 | 50.3 | 29.5 | 55.1 | 37.2 | 74.7 | 41.2 | 46.4 |
| BN Li & Hoiem (2017) | 42.1 | 40.7 | 42.7 | 27.6 | 41.9 | 29.7 | 27.9 | 34.9 | 35.0 | 41.5 | 26.5 | 30.3 | 35.7 | 32.9 | 41.2 | 35.4 |
| TENT Wang et al. (2020) | 37.2 | 35.8 | 41.7 | 37.9 | 51.2 | 48.3 | 48.5 | 58.4 | 63.7 | 71.1 | 70.4 | 82.3 | 88.0 | 88.5 | 90.4 | 60.9 |
| CoTTA Wang et al. (2022) | 40.1 | 37.7 | 39.7 | 26.9 | 38.0 | 27.9 | 26.4 | 32.8 | 31.8 | 40.3 | 24.7 | 26.9 | 32.5 | 28.3 | 33.5 | 32.5 |
| RoTTA Yuan et al. (2023) | 49.1 | 44.9 | 45.5 | 30.2 | 42.2 | 29.5 | 26.1 | 32.2 | 30.7 | 37.5 | 24.7 | 29.1 | 32.6 | 30.4 | 36.7 | 34.8 |
| RMT Döbler et al. (2023) | 40.2 | 36.2 | 36.0 | 27.9 | **33.9** | 28.4 | 26.4 | **28.7** | 28.8 | 31.1 | 25.5 | 27.1 | **28.0** | 26.6 | **29.0** | 30.2 |
| PETAL Brahma & Rai (2023) | 38.3 | 36.4 | 38.6 | **25.9** | 36.8 | **27.3** | 25.4 | 32.0 | 30.8 | 38.7 | 24.4 | 26.4 | 31.5 | 26.9 | 32.5 | 31.5 |
| DSS Wang et al. (2024) | 39.7 | 36.0 | 37.2 | 26.3 | 35.6 | 27.5 | **25.1** | 31.4 | 30.0 | 37.8 | 24.2 | **26.0** | 30.0 | 26.3 | 31.1 | 30.9 |
| SATA Chakrabarty et al. (2023) | **36.5** | **33.1** | **35.1** | **25.9** | 34.9 | 27.7 | 25.4 | 29.5 | 29.9 | 33.1 | **23.6** | 26.7 | 31.9 | 27.5 | 35.2 | 30.3 |
| Ours | 38.1 | 34.8 | 36.4 | 27.1 | 34.3 | 27.7 | 26.1 | **28.7** | **28.5** | **30.9** | 24.1 | 26.2 | 28.2 | **26.2** | 31.2 | **29.9** |

et al., 2020), CoTTA (Wang et al., 2022), RoTTA (Yuan et al., 2023), SATA (Chakrabarty et al., 2023), SWA (Yang et al., 2023), PETAL (Brahma & Rai, 2023), RMT (Döbler et al., 2023), DSS (Wang et al., 2024). All compared methods utilize the same backbone and pretrained model. All experiments are conducted on a single RTX 4090.

## 4.2 Major Results for Continual Test-Time Adaptation Benchmarks

**Experiments on CIFAR10-C.** We first evaluate the effectiveness of the proposed model on the CIFAR10-C dataset. We compare our method to the source-only baseline and nine SOTA methods. As shown in Table 1, directly using pre-trained model without adaptation yields a high average error rate of 43.5%. BN method improve the performance by 23.1% compared to the source-only baseline. Among all comparison methods, SWA achieve the lowest error rate of 11.2%, 11.8% and 11.5% on Defoc., Motion and Fog, respectively. Both PETAL and SATA achieve the lowest error rate of 7.4% on Brit.. In other conditions, our proposed method outperforms or is comparable to all the above methods. In conclusion, our method achieve the lowest average error rate, which is reduced to 14.7%.

**Experiments on CIFAR100-C.** To further demonstrate the effectiveness of the proposed method, we evaluate it on the more difficult CIFAR100-C task with the source-only baseline and eight SOTA methods. The experimental results are shown in Table 2. Generally speaking, our method not only achieved the lowest error rates on the Snow, Frost, Fog, and Pix. tasks but also had the lowest average error rate. We improve the performance by 16.5% and 0.4% compared to the source-only baseline and SATA, respectively.

**Experiments on ImageNet-C.** The last experiment is conducted on ImageNet-C to further demonstrate the effectiveness of the proposed method. The experimental results can be seen in Table 3. Compared with the SOTA methods, the proposed method achieve the lowest average error rate. Noteworthily, the proposed method outperforms SATA by a large margin for the Shot(71.6% vs. 72.9%), Impul. (68.7% vs. 71.6%), Defoc. (74.0% vs. 75.7%) and Glass. (71.6% vs. 74.1%) corruptions.

## 4.3 Ablation Studies

We perform ablation study experiments to evaluate the effectiveness of major components of C-CoTTA on three benchmarks. For simplicity, we denote the Class Shift Controlling Loss as CSCL

Table 3: Classification error rate (%) for standard ImageNet-C continual test-time adaptation task.

| Method | Gauss. | Shot | Impul. | Defoc. | Glass | Motion | Zoom | Snow | Frost | Fog | Brit. | Contr. | Elas. | Pix. | Jpeg | Mean |
|---|---|---|---|---|---|---|---|---|---|---|---|---|---|---|---|---|
| Source | 95.3 | 95.0 | 95.3 | 86.1 | 91.9 | 87.4 | 77.9 | 85.1 | 79.9 | 79.0 | 45.4 | 96.2 | 86.6 | 77.5 | 66.1 | 83.0 |
| BN Li & Hoiem (2017) | 87.7 | 87.4 | 87.8 | 88.0 | 87.7 | 78.3 | 63.9 | 67.4 | 70.3 | 54.7 | 36.4 | 88.7 | 58.0 | 56.6 | 67.0 | 72.0 |
| TENT Wang et al. (2020) | 81.6 | 74.6 | 72.7 | 77.6 | 73.8 | 65.5 | **55.3** | 61.6 | 63.0 | 51.7 | 38.2 | 72.1 | 50.8 | 47.4 | 53.3 | 62.6 |
| CoTTA Wang et al. (2022) | 84.7 | 82.1 | 80.6 | 81.3 | 79.0 | 68.6 | 57.5 | 60.3 | 60.5 | 48.3 | 36.6 | 66.1 | **47.2** | **41.2** | 46.0 | 62.7 |
| RoTTA Yuan et al. (2023) | 88.3 | 82.8 | 82.1 | 91.3 | 83.7 | 72.9 | 59.4 | 66.2 | 64.3 | 53.3 | **35.6** | 74.5 | 54.3 | 48.2 | 52.6 | 67.3 |
| RMT Döbler et al. (2023) | 79.9 | 76.3 | 73.1 | 75.7 | 72.9 | 64.7 | 56.8 | 56.4 | **58.3** | 49.0 | 40.6 | **58.2** | 47.8 | 43.7 | **44.8** | 59.9 |
| PETAL Brahma & Rai (2023) | 87.4 | 85.8 | 84.4 | 85.0 | 83.9 | 74.4 | 63.1 | 63.5 | 64.0 | 52.4 | 40.0 | 74.0 | 51.7 | 45.2 | 51.0 | 67.1 |
| DSS Wang et al. (2024) | 84.6 | 80.4 | 78.7 | 83.9 | 79.8 | 74.9 | 62.9 | 62.8 | 62.9 | 49.7 | 37.4 | 71.0 | 49.5 | 42.9 | 48.2 | 64.6 |
| ViDA Liu et al. (2023) | 79.3 | 74.7 | 73.1 | 76.9 | 74.5 | 65.0 | 56.4 | 59.8 | 62.6 | 49.6 | 38.2 | 66.8 | 49.6 | 43.1 | 46.2 | 61.2 |
| SATA Chakrabarty et al. (2023) | **74.1** | 72.9 | 71.6 | 75.7 | 74.1 | **64.2** | 55.5 | **55.6** | 62.9 | **46.6** | 36.1 | 69.9 | 50.6 | 44.3 | 48.5 | 60.1 |
| Ours | 75.1 | **71.6** | **68.7** | **74.0** | **71.6** | 65.1 | 56.4 | 55.7 | 61.0 | 49.3 | 41.3 | 61.9 | 49.1 | 44.8 | 46.0 | **59.4** |

Table 4: Ablation study on class and domain under severity 5.

| No. | CSCL | DSCL | CIFAR10-C | CIFAR100-C | ImageNet-C |
|---|---|---|---|---|---|
| 1 | | | 15.66 | 31.19 | 60.56 |
| 2 | ✓ | | 14.94 | 30.42 | 59.67 |
| 3 | | ✓ | 15.16 | 30.68 | 59.75 |
| 4 | ✓ | ✓ | 14.71 | 29.92 | 59.43 |

and Domain Shift Controlling Loss as DSCL. As shown in Table 4, C-CoTTA decrease error rates in all benchmarks after adding CSCL or DSCL to the network, which indicate mitigating inter-category interference or reducing the model's sensitivity to overall domain shifts can improve the classification accuracy. Furthermore, the combination of CSCL and DSCL can further improve the classification accuracy of the C-CoTTA framework.

## 4.4 t-SNE Visualization for Class Shift

We use t-SNE Van der Maaten & Hinton (2008) for dimensionality reduction to visualize domain shift situations of different methods during test-time. As shown in Figure 3, it can be observed that directly using pre-trained model (Source), CoTTA, and SATA methods do not explicitly control domain shift, resulting in unpredictable domain shift directions, leading to blurry classification boundaries and mutual interference between categories. In contrast, our method implements controllable domain shift, so it can be seen that although categories also experience shift, the direction of the shift is benign and does not cause mutual interference between categories.

## 4.5 Inter-Class Distance and Inter-Domain Distance

**Inter-Class Distance.** The inter-class distance can be formulate as $d_{ic} = \sum_{i=1}^{c} \sum_{j \neq i}^{c} \left\| \mathbf{p}_i^t - \mathbf{p}_j^t \right\|_2^2$, where $\mathbf{p}_i^t$ and $\mathbf{p}_j^t$ denote target domain category prototype. The inter-class distance comparison between our method, CoTTA and SATA can be seen in Figure 4(a). Compared to CoTTA and SATA, our method has a larger inter-class distance, indicating better separability between classes and reflecting the effectiveness of controlling class shift direction.

**Inter-Domain Distance.** The inter-domain distance can be computed as $d_{id} = \left\| \mathbf{p}^s - \mathbf{p}^t \right\|_2^2$, where $\mathbf{p}^s$ denotes overall prototype of the source domain while $\mathbf{p}^t$ denotes overall prototype of the target domain. As shown in the Figure 4(b), compared to SATA, our method has smaller inter-domain distances, indicating that the model is less sensitive to domain transformations and reflecting the effectiveness of controlling domain shift. At the same time, although the CoTTA method has relatively small inter-domain distances in the early stage, as the target domain changes at test time, its inter-domain distances gradually increase. This indicates that the model is sensitive to domain shift, further emphasizing the importance of reducing the sensitivity of the model to domain shift in controlling overall domain shift.

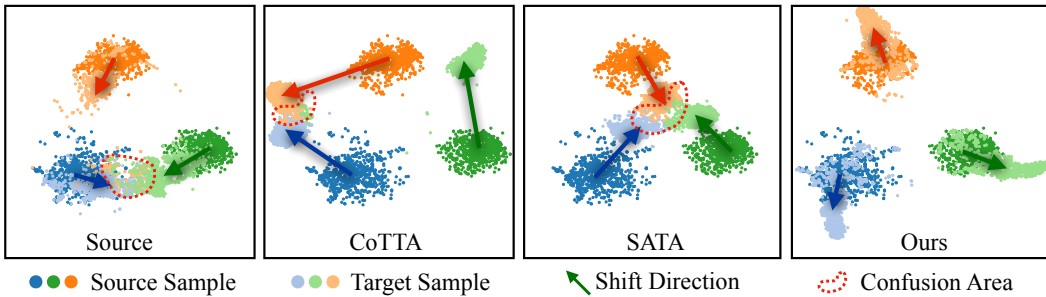

Figure 3: Visualization of the t-SNE dimensionality reduction of three classes from CIFAR10-C dataset (three easily misclassified animals: bird, deer, frog) transferred from the source domain to the target domain (zoom)

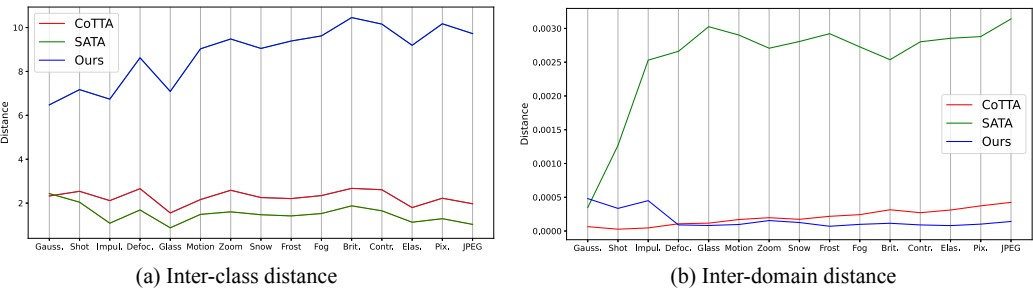

(a) Inter-class distance         (b) Inter-domain distance

Figure 4: (a) Inter-class distance: can indicate the separability between classes. (b) Inter-domain distance: can indicate the sensitivity of the model to domain transformations.

### 4.6 RESULTS FOR GRADUAL TEST-TIME ADAPTATION

In the standard setup described above, corruption types change abruptly at the highest severity level. We will now present the results of the gradual setup. We design the sequence by gradually changing the severity for the 15 types of corruption. When the type of corruption changes, the severity level is at its lowest. The distribution shift within each type is also gradual. Table 5 shows that the proposed method outperforms competing methods.

Table 5: Results for Gradual Adaptation

| Methods | CIFAR10-C | CIFAR100-C | ImageNet-C |
|---|---|---|---|
| Source | 23.9 | 32.9 | 81.7 |
| TENT | 39.1 | 72.7 | 53.7 |
| CoTTA | 10.6 | 26.3 | 42.1 |
| SATA | 10.8 | 27.5 | 44.8 |
| Ours | **9.21** | **26.2** | **41.1** |

Table 6: Results for Corruption loops Adaptation

| Methods | CIFAR10-C | CIFAR100-C | ImageNet-C |
|---|---|---|---|
| Source | 43.5 | 46.4 | 83.0 |
| TENT | 41.8 | 31.2 | 65.3 |
| CoTTA | 15.7 | 32.4 | 68.2 |
| SATA | 15.5 | 32.2 | 62.5 |
| Ours | **11.6** | **27.2** | **52.6** |

### 4.7 RESULTS FOR CORRUPTION LOOPS TEST-TIME ADAPTATION

In real-world scenarios, test domains may occur in cycles. To assess the long-term adaptation performance of the method, we evaluated the test conditions for 10 consecutive cycles. This means that at level 5 severity, the test data will be reanalyzed and readjusted nine times. The complete result can be found in Table 6. The results also show that our method outperforms the other methods in this long-term adaptation scenario. It illustrates the effectiveness of domain offset controllability as well as category offset controllability.

## 4.8 RESULTS FOR RANDOM ORDER TEST-TIME ADAPTATION

For a more comprehensive evaluation of the proposed method, CIFAR10-C, CIFAR100-C, and ImageNet-C experiments are conducted on over ten sequences of various corruption types with a severity level of 5. As shown in the Table 7, C-CoTTA is consistently outperforming CoTTA and other competing methods. This shows C-CoTTA is robust when facing different corruption orders.

Table 7: Average error of standard ImageNet-C experiments over 10 diverse corruption sequences.

| Avg. Error (%) | Source | TENT | CoTTA | SATA | Ours |
|---|---|---|---|---|---|
| CIFAR10-C | 43.5 | 20.1 | 16.3 | 16.3 | **14.7** |
| CIFAR100-C | 46.4 | 61.3 | 32.6 | 32.8 | **29.5** |
| ImageNet-C | 83.0 | 61.8 | 57.9 | 64.5 | **55.5** |

## 4.9 SEMANTIC SEGMENTATION ON CITYSCAPES-TO-ACDC

We further evaluate our methodology in the context of the more practical continual test-time semantic segmentation task. We conduct our experiments on Cityscapes-to-ACDC dataset, and use ViT (Segformer-B5) as backbone. The results are shown in Table 8. The results indicate that our approach is not only effective for semantic segmentation tasks but also demonstrates robustness across various architectural configurations. Our proposed method achieves an absolute improvement of 0.6% in mean Intersection over Union (mIoU) compared to the baseline, resulting in a total mIoU of 59.2%. It is noteworthy that existing methods such as BN and TENT exhibit suboptimal performance in this task, with a marked decline in efficacy over time.

Table 8: Semantic segmentation results (mIoU in %) on the Cityscapes-to-ACDC Sakaridis et al. (2021) online continual test-time adaptation task. We evaluate the four test conditions continually for ten times to evaluate the long-term adaptation performance. To save space, we only show the continual adaptation results in the first, fourth, seventh, and last round. Full results can be found in the supplementary material. All results are evaluated based on the Segformer-B5 Xie et al. (2021) architecture.

| Time | $t$ | | | | | | | | | | | | | | | | |
|---|---|---|---|---|---|---|---|---|---|---|---|---|---|---|---|---|---|
| Round | | 1 | | | | 4 | | | | 7 | | | | 10 | | | |
| Condition | Fog | Night | rain | snow | Fog | Night | rain | snow | Fog | Night | rain | snow | Fog | Night | rain | snow | Mean |
| Source | 69.1 | 40.3 | 59.7 | 57.8 | 69.1 | 40.3 | 59.7 | 57.8 | 69.1 | 40.3 | 59.7 | 57.8 | 69.1 | 40.3 | 59.7 | 57.8 | 56.7 |
| BN | 62.3 | 38.0 | 54.6 | 53.0 | 62.3 | 38.0 | 54.6 | 53.0 | 62.3 | 38.0 | 54.6 | 53.0 | 62.3 | 38.0 | 54.6 | 53.0 | 52.0 |
| TENT | 69.0 | 40.2 | 60.1 | 57.3 | 66.5 | 36.3 | 58.7 | 54.0 | 64.2 | 32.8 | 55.3 | 50.9 | 61.8 | 29.8 | 51.9 | 47.8 | 52.3 |
| CoTTA | 70.9 | 41.2 | 62.4 | 59.7 | 70.9 | 41.0 | 62.7 | 59.7 | 70.9 | 41.0 | 62.8 | 59.7 | 70.8 | 41.0 | 62.8 | 59.7 | 58.6 |
| Ours | **72.3** | **42.2** | **62.8** | **59.9** | **71.9** | **41.2** | **63.7** | **60.2** | **71.2** | **42.0** | **63.2** | **60.3** | **71.4** | **42.1** | **62.9** | **59.9** | **59.2** |

## 5 CONCLUSION AND LIMITATION

In this work, we introduce C-CoTTA, a novel framework designed to prevent any category from leaning towards other categories by explicitly controlling the offset direction to avoid fuzzy classification boundaries, and reduce the sensitivity of the model in the domain shift direction to reduce the impact of domain shift on domain adaptation. This fills the gap left by traditional methods, which can only mitigate the impact of domain drift. C-CoTTA can explicitly control domain shift, opening up a new solution pathway for CTTA. Through extensive quantitative experiments and qualitative analysis, such as t-SNE plots, we demonstrate the effectiveness and theoretical validity of C-CoTTA.

Our method also has certain limitations. During test-time, we may not have access to all samples of a specific domain prototype, and due to the lack of true labels, misclassified samples may contaminate the prototype. As a result, the representation of the prototype may be poor, further affecting the accuracy of the constructed direction, leading to ineffective or even erroneous domain shift control. Therefore, in the future, addressing how to obtain high-quality prototypes and directional representations is a task that needs attention.

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

## A EQUIVALENT REPRESENTATION OF SCAV

Further derivation of scav is conducted to obtain a simpler equivalent representation, as follows:

$$
\begin{aligned}
\mathrm{cov}[f(x), y] &= \sum_{x_i \in \mathcal{X}_c \cup \mathcal{X}_n} (f(x_i) - \overline{f}(x))(y_i - \overline{y}) \\
&= \frac{|\mathcal{X}_n|}{|\mathcal{X}_c| + |\mathcal{X}_n|} \sum_{x_i \in \mathcal{X}_c} (f(x_i) - \overline{f}(x)) + \frac{|\mathcal{X}_c|}{|\mathcal{X}_c| + |\mathcal{X}_n|} \sum_{x_i \in \mathcal{X}_n} (f(x_i) - \overline{f}(x)) \\
&= \frac{|\mathcal{X}_n|}{|\mathcal{X}_c| + |\mathcal{X}_n|} \left( \sum_{x_i \in \mathcal{X}_c} f(x_i) - |\mathcal{X}_c| \cdot \overline{f}(x_i) \right) - \frac{|\mathcal{X}_c|}{|\mathcal{X}_c| + |\mathcal{X}_n|} \left( \sum_{x_i \in \mathcal{X}_n} f(x_i) - |\mathcal{X}_n| \overline{f}(x_i) \right) \\
&= \frac{|\mathcal{X}_n|}{|\mathcal{X}_c| + |\mathcal{X}_n|} \sum_{x_i \in \mathcal{X}_c} f(x_i) - \frac{|\mathcal{X}_n||\mathcal{X}_c|}{|\mathcal{X}_c| + |\mathcal{X}_n|} \overline{f}(x_i) - \frac{|\mathcal{X}_c|}{|\mathcal{X}_c| + |\mathcal{X}_n|} \sum_{x_i \in \mathcal{X}_n} f(x_i) + \frac{|\mathcal{X}_c||\mathcal{X}_n|}{|\mathcal{X}_c| + |\mathcal{X}_n|} \overline{f}(x_i) \\
&= \frac{|\mathcal{X}_n||\mathcal{X}_c|}{|\mathcal{X}_c| + |\mathcal{X}_n|} \left( \frac{1}{|\mathcal{X}_c|} \sum_{x_i \in \mathcal{X}_c} f(x_i) - \frac{1}{|\mathcal{X}_n|} \sum_{x_i \in \mathcal{X}_n} f(x_i) \right) \\
\mathrm{cov}[y, y] &= \sum_{x_i \in \mathcal{X}_c \cup \mathcal{X}_n} (y_i - \overline{y})^2 \\
&= \sum_{x_i \in \mathcal{X}_c} (1 - \frac{|\mathcal{X}_c|}{|\mathcal{X}_c| + |\mathcal{X}_n|})^2 + \sum_{x_i \in \mathcal{X}_n} (0 - \frac{|\mathcal{X}_c|}{|\mathcal{X}_c| + |\mathcal{X}_n|})^2 \\
&= \sum_{x_i \in \mathcal{X}_c} \frac{|\mathcal{X}_n|^2}{(|\mathcal{X}_c| + |\mathcal{X}_n|)^2} + \sum_{x_i \in \mathcal{X}_n} \frac{|\mathcal{X}_c|^2}{(|\mathcal{X}_c| + |\mathcal{X}_n|)^2} \\
&= \frac{|\mathcal{X}_c||\mathcal{X}_n|^2 + |\mathcal{X}_n||\mathcal{X}_c|^2}{(|\mathcal{X}_c + |\mathcal{X}_n|)^2} \\
&= \frac{|\mathcal{X}_c||\mathcal{X}_n|}{|\mathcal{X}_c| + |X_n|} \\
\frac{\mathrm{cov}[f(x), y]}{\mathrm{cov}[y, y]} &= \frac{1}{|\mathcal{X}_c|} \sum_{x_i \in \mathcal{X}_c} f(x_i) - \frac{1}{|\mathcal{X}_n|} \sum_{x_i \in \mathcal{X}_n} f(x_i)
\end{aligned}
$$

## B CORRUPTION LOOPS TEST-TIME ADAPTATION

We present the results of 10 cycles on CIFAR10-C. As depicted in the Fig. 5, it is evident that over time, the error rate of CoTTA and SATA methods has gradually increased, whereas our method continues to decrease. Consequently, the performance gap is widening. This phenomenon is most likely related to the fact that the first two methods lack reasonable control over the category and the offset direction of the domain.

## C HYPERPARAMETER ANALYSIS

In this section, we delve into the critical examination of hyperparameters $\lambda_1$ and $\lambda_2$ in Eq. 11 within our C-CoTTA framework in ImageNet-C, which substantially influence the model's performance. Through meticulous experimentation, we fine-tuned these hyperparameters to identify their optimal values, ensuring the harmonious interplay between Domain Shift Controlling Loss (DSCL) and Class Shift Controlling Loss (CSCL) components.

Our investigation revealed that the selection of $\lambda_1$ and $\lambda_2$ is pivotal in balancing the contributions of DSCL and CSCL to the overall objective function. We experimented with a spectrum of values for these hyperparameters, meticulously recording the impact on classification accuracy and domain adaptation efficacy. The empirical results, illustrated in Fig. 6, present a compelling case for the

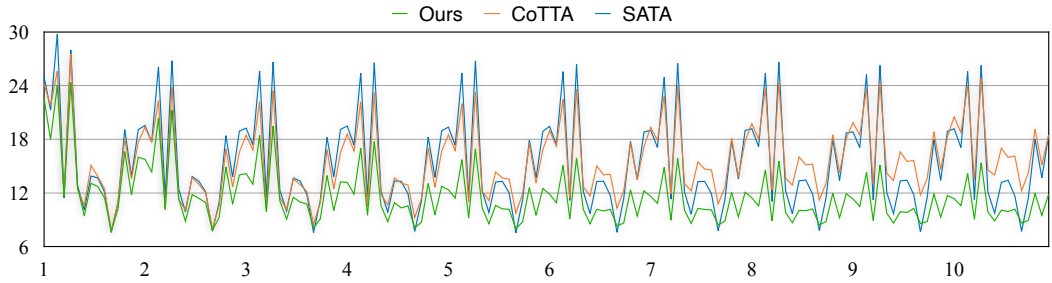

Figure 5: Under the CIFAR10-C dataset, it can be observed that the performance of each method under 10 corruption cycles varies. The error rates of CoTTA and SATA methods start to gradually increase in the later stages, whereas our method continues to decrease or maintain stability.

optimal balance that our chosen hyperparameters provide, underscoring the model's robustness against various disturbances.

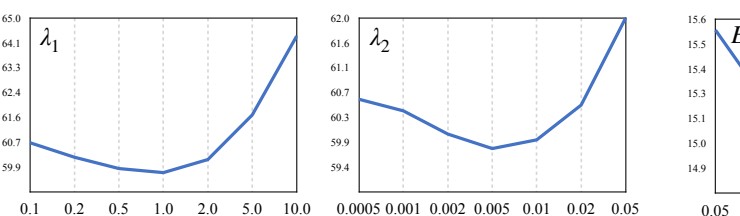

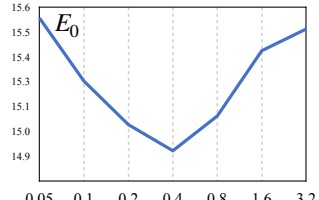

Figure 6: Analysis Hyperparameter of $\lambda_1$ for DSCL and $\lambda_2$ for CSCL on ImageNet-C.

Figure 7: Analysis Hyperparameter of $E_0$ on CIFAR10-C.

## D  RELIABLE SAMPLE SELECTION ANALYSIS

In our control category offset method, in order to reduce the contamination of category prototypes by misclassified samples and thus affect the control of category shift direction, we remove samples with entropy values exceeding the predefined threshold $E_0$, which is set as $0.4 \times \ln C$ based on Niu et al. (2022). We have verified the rationality of this operation through experiments. As shown in Fig. 7, when the threshold is large, the effectiveness of the CSCL method deteriorates. This may be because the conditions are too loose, leading to a large number of misclassified samples when calculating the prototype. On the other hand, when the threshold is small, the CSCL method also deteriorates. This may be because the conditions are too strict, resulting in too few samples used to calculate the prototype, making the generated prototype not representative.

## E  CLASS CONFUSION MATRIX

We observed the confusion matrix of category in the domain adaptation process. As shown in Fig. 8, compared to the CoTTA and SATA methods, our method significantly reduced the degree of category confusion, demonstrating the effectiveness of controllable domain shift.

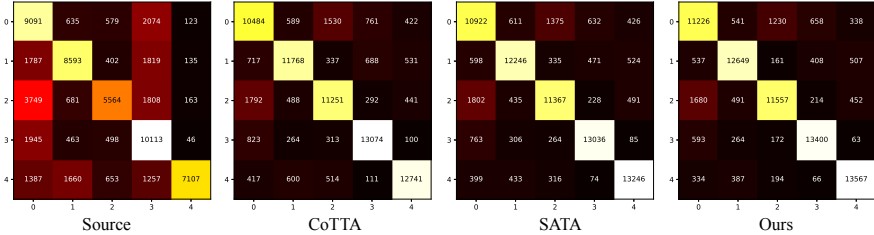

Figure 8: confusion matrix of category in the domain adaptation process. The vertical axis represents the true labels, and the horizontal axis represents the predicted labels.

