# OpenReview forum: "Controllable Continual Test-Time Adaptation"
_ICLR.cc/2025/Conference — ICLR 2025 Conference Withdrawn Submission_

### Official Review · Reviewer_FZMn · 2024-10-28

**Soundness:** 3
**Presentation:** 1
**Contribution:** 3
**Rating:** 3
**Confidence:** 5

**Summary:**

This paper proposes Controllable Continual Test-Time Adaptation (C-CoTTA), a method that enables models to adapt to continuously changing domains during test time without access to the original source data.

Unlike traditional CTTA approaches, which primarily focus on suppressing domain shifts to mitigate error accumulation, this approach controls the direction of domain shifts to prevent category confusion. By leveraging Concept Activation Vectors (CAV), the method controls the feature shift direction for each class, while domain- and class-level shift control loss functions enhance classification performance.

**Strengths:**

The strength of this paper lies in introducing a new loss function to actively control domain shifts within the CTTA framework. This approach not only tackles error accumulation but also provides a way to maintain clear boundaries between classes during continual adaptation.

**Weaknesses:**

1. There are a lot of grammatical errors that hinder readers from concentrating on the paper.
For example

- Line 109: focuses ⇒ focus

- Line 112: they are designed ⇒ they designed

- Line 124: Andres et al. (2022). no parenthesis in the reference.

- Line 157: refer to ⇒ refers to

- Line 225: calculates ⇒ is calculated

- Line 231: weird sentence

- etc.

Also, there are quite a lot of notational errors

- eq 2: $\mathcal{X}_t$ and $\mathcal{X}_s$ are undefined

- eq 3 and 4: inconsistent notation (both subscripts and superscripts are used to indicate the same objects)

- Line 216: $i$ is undefined

Poorly structured English and inconsistent symbols can indeed make the core concepts harder to follow, even if the underlying ideas are strong. Improved clarity in language and notation would make it easier for readers to fully appreciate the novel approach to CTTA and better understand the implementation of the new loss functions for domain control.

I recommend that the authors thoroughly revise the grammar and notation before resubmitting the paper to a new venue.

2.The idea of using a domain prototype was originally proposed in the UDA task, so it’s not a new concept. However, it has not been actively utilized in TTA because it’s challenging to obtain an accurate domain prototype given the nature of TTA, which receives online mini-batches as input. What is the mini-batch size used in this experiment? Mini-batch size is considered a significant issue in TTA.

3. It still inherits the weaknesses of previous works that use pseudo labels. Depending on the accuracy of the pseudo labels, there is a possibility that the class shift may be inaccurately estimated.

**Questions:**

1. Why the symmetric cross-entropy loss is used rather than KL or reverse KL in eq. 10?

2. What happens if a batch contains samples from multiple domains? I wonder how much performance degradation would result from that scenario.

3. Figure 3 is quite intuitive. Can you show similar figures for different target domains over time?

4. The performance of RMT recorded in this paper is lower than the results reported in the original RMT paper, with a noticeable difference. What is the reason for this?

---

> ### Author Response · Authors · 2024-11-15
>
> Thank you for your valuable comments and kind words to our work. At the same time, we sincerely apologize for any inconvenience caused by some grammatical and notational errors in your reading. The revised article has been re-uploaded. Below we address specific questions.
>
> **Q1. Why the symmetric cross-entropy loss is used rather than KL or reverse KL in eq. 10?**
>
> A1. SCE loss is based on previous work RMT, where they found that SCE maintains a balance in the gradient for high and low confidence predictions, which benefits the optimization problem.
>
> **Q2. What happens if a batch contains samples from multiple domains? I wonder how much performance degradation would result from that scenario.**
>
> A2. The experimental setup you described is different from the CTTA (Continual Test-Time Adaptation) that we are studying; it is another setup called WTTA (Test-time Adaptation in Dynamic Wild World), which was first proposed by SAR$^1$. Due to the differences in experimental details, the performance of the two setups cannot be directly compared.
>
> **Q3. Figure 3 is quite intuitive. Can you show similar figures for different target domains over time?**
>
> A3. Of course, we have provided the results for an additional 4 domains (noise, blur, weather, digital) in the tsne_big.pdf in the Supplementary Material, please take a look.
>
> **Q4. The performance of RMT recorded in this paper is lower than the results reported in the original RMT paper, with a noticeable difference. What is the reason for this?**
>
> A4. Because the warm-up operation in RMT uses a large amount of source domain data, we believe this violates the setup of CTTA. Therefore, the data reported in our paper is after removing this operation. (If we also included this operation, the results would improve, but clearly, this is unreasonable.)
>
> **Q5. It still inherits the weaknesses of previous works that use pseudo labels. Depending on the accuracy of the pseudo labels, there is a possibility that the class shift may be inaccurately estimated.**
>
> A5. This is indeed an important issue, which has been mentioned in the summary section of our paper (line 288). At the same time, we have made significant efforts to address it (line 145). We adopted the method from EATA by setting an entropy threshold to filter out samples with low entropy, which indicates higher uncertainty, and used the certain good samples to construct prototypes.
>
> **Q6. The idea of using a domain prototype was originally proposed in the UDA task, so it’s not a new concept. However, it has not been actively utilized in TTA because it’s challenging to obtain an accurate domain prototype given the nature of TTA, which receives online mini-batches as input. What is the mini-batch size used in this experiment? Mini-batch size is considered a significant issue in TTA.**
>
> A6. In terms of setting the batch size, we follow the previous methods CoTTA and RMT. Specifically, the batch sizes for cifar10-c and cifar100-c are set to 200, while for imagenet-c it is set to 64. At the same time, we also recognize that mini-batch size is considered a significant issue, especially having a considerable impact on the acquisition of class prototypes. Therefore, we conducted tests on different batch sizes for imagenet-c, and the results are as follows:
>
> |       | 64   | 32   | 16   | 8    | 4    |
> | ----- | ---- | ---- | ---- | ---- | ---- |
> | Ours  | 59.4 | 60.0 | 62.4 | 68.3 | 76.5 |
> | SATA  | 60.1 | 62.1 | 64.0 | 72.4 | 84.2 |
> | RMT   | 59.9 | 62.1 | 65.5 | 71.8 | 83.8 |
> | CoTTA | 62.7 | 63.1 | 64.7 | 70.5 | 78.4 |
>
> If there are any parts of my response that still confuse you, please point them out, and I will reply promptly.
>
> [1] Niu, Shuaicheng, et al. "Towards stable test-time adaptation in dynamic wild world." arXiv preprint arXiv:2302.12400 (2023).

---

> > ### Author Response · Authors · 2024-12-03
> >
> > Thanks again for your insightful comments and valuable time devoted to our paper. As the author-reviewer discussion period is coming to an end, please let us know if there's any further information we can provide to facilitate the discussion process. We are happy to answer any questions or concerns you may have during the author-reviewer discussion period.

---

> > > ### Comment · Reviewer_FZMn · 2024-12-03
> > > **I keep my original score.**
> > >
> > > Due to a lack of readability, I keep my original rating.
> > >
> > > I recommend that the authors thoroughly revise the paper before resubmitting the paper to a new venue.

---

### Official Review · Reviewer_VBFJ · 2024-10-30

**Soundness:** 3
**Presentation:** 3
**Contribution:** 2
**Rating:** 3
**Confidence:** 4

**Summary:**

The paper presents a novel framework for continual test-time adaptation (CTTA), aiming to control domain shifts dynamically during model inference without access to source data or labels. The approach, named C-CoTTA, introduces mechanisms to guide domain shifts to maintain clear class boundaries and minimize error accumulation.

**Strengths:**

1. The paper addresses a significant challenge in CTTA by proposing a method to control domain shifts, which is crucial for applications in dynamic environments.
2. The approach uses Concept Activation Vectors (CAVs) to represent and control shift directions, which is a well-founded technique in interpretable AI.

**Weaknesses:**

1. Marginal Improvements: The reported improvements in classification accuracy are relatively marginal (0.6%, 0.4%, 0.5% on CIFAR-10C, CIFAR100-C, and ImageNet-C, respectively). This raises concerns about the practical significance and robustness of the proposed method.
2. Inconsistency in Compared Methods: There is a lack of consistency in the methods compared across different datasets. For instance, ViDA is only included in the ImageNet-C experiments but not in CIFAR experiments. Additionally, the paper does not include comparisons with more recent and potentially more effective methods in segmentation tasks.
3. Metric Validity and Relevance Concerns: The paper uses inter-class and inter-domain distances as metrics to assess class separability and sensitivity to domain shifts. However, the relevance and validity of these metrics are questionable. For instance, the paper does not account for the potential scaling of feature values; if feature vectors $p_i^t$ and $p_j^j$ in line 414 are simply scaled up by a factor (e.g., doubled), the computed distances would also increase, suggesting greater separability without actual improvement in class distinction. This is illustrated by the example where different domains with similar inter-class and inter-domain distances exhibit significant accuracy disparities, such as between the 'Brightness' and 'Contrast' conditions of ImageNet-C, which have a roughly 20% gap in accuracy despite similar distance metrics.
4. The paper does not discuss several relevant works such as [1-5]. This comparison is crucial for situating the novelty of the proposed method within the existing literature.
5. Given the use of labeled source data in the adaptation process, the experiments may not offer a fair comparison to results from methods that do not use labeled source data, rely only on statistics of source data, or do not use source data at all. Clarifying the conditions under which each method is evaluated is essential for understanding and interpreting the experimental results accurately.

Addressing these concerns could significantly strengthen the manuscript, enhancing its contribution to the field and its potential for a higher rating.

[1] Lee et al., "Becotta: Input-dependent online blending of experts for continual test-time adaptation", ICML 2024.

[2] Song et al., "Ecotta: Memory-efficient continual test-time adaptation via self-distilled regularization", CVPR 2023.

[3] Gong et al., "SoTTA: Robust Test-Time Adaptation on Noisy Data Streams", NeurIPS 2023.

[4] Liu et al., "Continual-MAE: Adaptive Distribution Masked Autoencoders for Continual Test-Time Adaptation", CVPR 2024.

[5] Yang et al., "A Versatile Framework for Continual Test-Time Domain Adaptation: Balancing Discriminability and Generalizability". CVPR 2024.

**Questions:**

1. Could you clarify how "random order" is defined and implemented in the experiments mentioned in section 4.8?
2. Does the availability of source data impact the performance of your method? Testing with varying amounts of source data could provide insights into its practical utility when labeled source data is limited.
3. The computation symbol "cov[]" used in formula (1) is unclear. Could you provide a definition or explanation?
4. Can you elaborate on the relationship between formulas (7) and (8)? What is the underlying intuition linking these formulas?
5. Considering practical constraints where not all classes may appear in each batch (e.g., batch sizes less than 1000), how might this affect the calculation of domain shift direction and the applicability of formula (9)?
6. Typo in lines 71 and 73. DSCL $\rightarrow$ CSCL?

---

> ### Author Response · Authors · 2024-11-15
>
> Thank you for your valuable comments and kind words to our work. Below we address specific questions.
>
> **Q1. Could you clarify how "random order" is defined and implemented in the experiments mentioned in section 4.8?**
>
> A1. In the CTTA experiment, there are 15 target domains, and everyone tests them in the same order (i.e., the order in Table 123). This may lead to the method overfitting to this order, so we randomly selected 10 different domain variation orders for testing, and the final results are averaged.
>
> **Q2. Does the availability of source data impact the performance of your method? Testing with varying amounts of source data could provide insights into its practical utility when labeled source data is limited.**
>
> A2. Note that during the test time, we only used the class prototypes extracted from the source domain data and did not use all the source domain data.
>
> **Q3. The computation symbol "cov[]" used in formula (1) is unclear. Could you provide a definition or explanation?**
>
> A3. In our article, "cov[]" means covariance.
>
> **Q4. Can you elaborate on the relationship between formulas (7) and (8)? What is the underlying intuition linking these formulas?**
>
> A4. In brief, formula (8) is the objective of our loss, while formula (7) is the implementation of our loss. We hope that the model's output is insensitive to changes in the domain, meaning that the model output should not change when activations are slightly added or removed along the domain bias direction, as stated in formula (8). We can achieve this goal by suppressing the directional derivative, as shown in formula (7).
>
> **Q5. Considering practical constraints where not all classes may appear in each batch (e.g., batch sizes less than 1000), how might this affect the calculation of domain shift direction and the applicability of formula (9)?**
>
> A5. We will only perform calculations on the categories that exist within the batch.
>
> **Q6. Typo in lines 71 and 73. DSCL $\to$ CSCL?**
>
> A6. Yes, this is a typo, and we will correct it.
>
> **Q7. Marginal Improvements**
>
> A7. The improvement in classification accuracy of the report is relatively small (Table 123), which may be related to the single order of the domain. Most of the previous work on CTTA adopted a single order, which may not reasonably assess the performance of various methods. Therefore, we supplemented with new experiments (section 4.8). We randomly selected 10 different domain variation orders for testing, and the final results are averaged. It can be seen that in this case, our improvement is significant (1.6%, 3.1%, and 2.4% on CIFAR-10C, CIFAR100-C, and ImageNet-C, respectively).
>
> **Q8. Inconsistency in Compared Methods**
>
> A8. ViDA was only included in the ImageNet-C experiments and not in the CIFAR experiments because we directly used the data provided in the ViDA paper. However, the backbone used in the ViDA paper for the cifar10-c and cifar100-c datasets is ViT, while we used ResNet, so there is no comparability. The same applies to other methods.
>
> If there are any parts of my response that still confuse you, please point them out, and I will reply promptly.

---

> > ### Comment · Reviewer_VBFJ · 2024-11-18
> >
> > > everyone tests them in the same order (i.e., the order in Table 123)
> >
> > Not ture. See Table 6 in CoTTA or Table 2 in EcoTTA. Still the number of baseline methods in Table 7 is much less than Table 123 if you want to compare C-CoTTA in diverse settings (Same for A7).
> >
> > > we only used the class prototypes extracted from the source domain data and did not use all the source domain data.
> >
> > So how does the performance of the proposed method change when class prototypes are extracted from less source data?
> >
> > > We will only perform calculations on the categories that exist within the batch.
> >
> > My question is will only computing with existing classes affect the overall performance across all classes?
> >
> > > However, the backbone used in the ViDA paper for the cifar10-c and cifar100-c datasets is ViT, while we used ResNet, so there is no comparability.
> >
> > I don't see why it is not comparable. ViDA was evaluated using ResNet50 on ImageNet-C (Table 1) and is obviously applicable to CIFAR. The absence of direct results in their paper is not a valid reason to exclude it from comparison.

---

> ### Author Response · Authors · 2024-11-23
>
> Thank you for the quick response.
>
> **Q1: Still the number of baseline methods in Table 7 is much less than Table 123 if you want to compare C-CoTTA in diverse settings. The absence of direct results in their paper is not a valid reason to exclude it from comparison.**
>
> A1: In order to better compare performance, we conducted experiments with the comparison methods on the CIFAR10-C, CIFAR100-C, and ImageNet-C datasets, using 10 different random corruption sequences. The backbone is the same for the same dataset. The table is as follows:  (Note that the two methods, ViDA and CMAE, both used ViT for experiments in the original paper on cifar10-c and cifar100-c. Here, for comparison, we use the same ResNet backbone as other methods.)
>
> |         | cifar10-c | cifar100-c | Imagenet-c |
> | ------- | --------- | ---------- | ---------- |
> | Source  | 43.5      | 46.4       | 83.0       |
> | Tent    | 20.1      | 61.3       | 61.8       |
> | CoTTA   | 16.3      | 32.6       | 57.9       |
> | SATA    | 16.3      | 32.8       | 64.5       |
> | RMT     | 16.2      | 32.8       | 63.8       |
> | DSS     | 17.5      | 33.4       | 67.5       |
> | SWA     | 15.8      | 31.5       | 60.9       |
> | PETAL   | 17.1      | 35.6       | 69.5       |
> | RoTTA   | 20.1      | 34.9       | 68.5       |
> | ViDA    | 15.1      | 30.5       | 57.3       |
> | Ecotta  | 16.8      | 33.8       | 62.5       |
> | Becotta | 15.8      | 31.6       | 58.9       |
> | CMAE    | 14.5      | 29.9       | 56.5       |
> | C-CoTTA | 14.7      | 29.5       | 55.5       |
>
>
> **Q2: how does the performance of the proposed method change when class prototypes are extracted from less source data.**
>
> A2: We conducted observations through experiments, taking the CIFAR100-C dataset as an example, where each category has 500 images. Previously, we used all 500 images to extract category prototypes. Now, we are testing the scenarios of using 1 image, 5 images, 10 images, 50 images, 100 images, and 500 images, to observe the impact of the number of source domain data on the method, as shown in the table below: （Note that the results for batch sizes ranging from 1 to 100 are obtained by running the experiment 5 times and taking the average, as the source domain data selected may vary each time.）
>
> |      | 1    | 5    | 10   | 50   | 100  | 500  |
> | ---- | ---- | ---- | ---- | ---- | ---- | ---- |
> | mean | 30.5 | 30.0 | 30.0 | 29.9 | 29.9 | 29.9 |
> | var  | 0.5  | 0.2  | 0.1  | 0.1  | 0.1  | \    |
>
>
>
>
>
> **Q3: will only computing with existing classes affect the overall performance across all classes.**
>
> A3:  Since the batch size in the experimental setup is smaller than the total number of categories, we cannot determine how the performance would be when optimizing with all categories.
>
> If there are any parts of my response that still confuse you, please point them out, and I will reply promptly.

---

> > ### Comment · Reviewer_VBFJ · 2024-11-25
> >
> > Thank you for providing the new results.
> >
> > I have two concerns about the first table. First, the authors repeatedly emphasize the model architecture. Is the proposed method only applicable to CNNs, or can it also be applied to other types of models, such as ViTs, which have been utilized in existing works? Second, the authors argued that the marginal improvement is due to only a single round. However, in the 10-round results, the improvement over SOTA appears to be not particularly significant (single-round: 0.6%, 0.4%, 0.5% vs. 10-round: -0.2%, 0.4%, 1.0%).
> >
> > While the second result demonstrates that C-CoTTA is robust against the number of source data, the concern about unfair comparisons remains. For example, Tent and CoTTA do not require any prototypes or information from labeled source data but C-CoTTA does

---

> > > ### Author Response · Authors · 2024-11-26
> > >
> > > Thank you for your interest in our work and your suggestions. Here is our answer to your confusion.
> > >
> > > **Q1: Is the proposed method only applicable to CNNs, or can it also be applied to other types of models, such as ViTs, which have been utilized in existing works?**
> > >
> > > A1: Of course, we conducted experiments on the CIFAR10-C, CIFAR100-C, and ImageNet-C datasets using the same ViT backbone as ViDA. The results are shown in the table below, and it can be seen that our method also performs well based on ViT.
> > >
> > > |      | cifar10-c | cifar100-c | imagenet-c |
> > > | ---- | --------- | ---------- | ---------- |
> > > | ViDA | 20.7      | 27.3       | 43.4       |
> > > | Ours | 19.8      | 25.5       | 39.9       |
> > >
> > > Note that we implemented our method within the code framework of the ViDA method, and aside from the parameters specific to our method, all other parameters remained unchanged to ensure fairness in comparison.
> > >
> > > **Q2: in the 10-round results, the improvement over SOTA appears to be not particularly significant.**
> > >
> > > A2: This is mainly because the performance of the CMAE method is better (we did not compare it in time with the method from CVPR 2024, so it was not included in the single-round experiments; instead, we followed your suggestion during the rebuttal phase and directly added it to the 10-round experiments). However, to test the effectiveness of our method, we added our two losses, CSCL and DSCL, based on CMAE (our two losses are plug-and-play). The results are shown in the table below, indicating that our method can further improve CTTA performance.
> > >
> > > |                | cifar10-c | cifar100-c | imagenet-c |
> > > | -------------- | --------- | ---------- | ---------- |
> > > | CMAE           | 14.5      | 29.9       | 56.5       |
> > > | CMAE+CSCL      | 14.1      | 29.1       | 55.5       |
> > > | CMAE+DSCL      | 14.3      | 29.4       | 56.1       |
> > > | CMAE+CSCL+DSCL | 13.9      | 28.8       | 55.2       |
> > >
> > > **Q3: While the second result demonstrates that C-CoTTA is robust against the number of source data, the concern about unfair comparisons remains. For example, Tent and CoTTA do not require any prototypes or information from labeled source data but C-CoTTA does**
> > >
> > > A3: Methods like Tent, CoTTA, and CMAE indeed do not use source prototypes, which can lead to certain unfairness. However, we believe that the original intention of CTTA is that test-time adaptation occurs on terminal devices (such as self-driving cars in motion). Due to storage capacity limitations and privacy concerns, it is necessary to avoid using all data from the source domain directly. However, using prototypes for each category does not impose a significant storage burden or privacy risk. Therefore, we believe that using prototypes in the CTTA scenario is reasonable, and RMT (CVPR 2023) also utilized source domain prototypes.
> > >
> > > If there are any parts of my response that still confuse you, please point them out, and I will reply promptly.

---

> > > > ### Comment · Reviewer_VBFJ · 2024-11-29
> > > >
> > > > > our two losses are plug-and-play
> > > >
> > > > Correct me if I'm wrong. The plug-and-play feature has not been mentioned anywhere in the manuscript. The proposed method is C-CoTTA, which is compared with other baselines. However, the authors suddenly claim this plug-and-play feature and combine their loss with another method (i.e., CMAE). This makes the contributions of the paper unclear and potentially confusing.
> > > >
> > > > > Due to storage capacity limitations and privacy concerns,
> > > >
> > > > Storage capacity is not my concern. My concern is that the improvement of C-CoTTA may stem from additional information obtained from labeled source data, compared with methods that do not assume access to any source data (e.g., Tent). If the higher improvement primarily results from stronger assumptions, it becomes difficult to argue that the proposed method is genuinely superior to other baselines.

---

> > > > > ### Author Response · Authors · 2024-11-30
> > > > >
> > > > > Thank you for your interest in our work and your suggestions. Here is our answer to your confusion.
> > > > >
> > > > > A1. First of all, it is certain that we did not indicate in the text that the two losses are plug-and-play.
> > > > >
> > > > > Secondly, the reason we introduced plug-and-play is that the CMAE method performs quite well and is very close to the performance of our method. Therefore, we wanted to test whether adding our method on top of CMAE would lead to an improvement in performance, in order to demonstrate the effectiveness of our method. At the same time, the two losses we proposed happen to be plug-and-play, so we conducted experiments by incorporating them into CMAE.
> > > > >
> > > > > A2. I agree with your point that the improvements of C-CoTTA may indeed stem from the additional information obtained from labeled source data (we acknowledge this, as our method cannot be executed without the source domain prototype). Therefore, it is unfair to compare it with methods that do not access any source data.
> > > > >
> > > > > However, both RMT and SATA methods utilize category prototypes (both use prototypes for contrastive learning), making them comparable, and our results outperform theirs.
> > > > >
> > > > > Nonetheless, we maintain that using prototypes does not fundamentally violate the setup of CTTA; rather, it is a reasonable way to utilize resources.
> > > > >
> > > > > If there are any parts of my response that still confuse you, please point them out, and I will reply promptly.

---

> ### Comment · Reviewer_VBFJ · 2024-12-03
>
> Thank you for the response. I appreciate the clarification provided, but some of my concerns remain unresolved. Specifically:
>
> 1. **Unclear Contributions:** The contributions of the paper are still not clear. If the plug-and-play feature is a key contribution, it needs to be validated with various existing CTTA methods to demonstrate its generalizability.
> 2. **Fairness of Comparison with Baselines:** The proposed method assumes stronger conditions compared to some baselines. A thorough discussion of the specific conditions under which C-CoTTA is compared to other methods would be helpful. [W5]
> 3. **Metric Validity and Relevance:** My concern regarding the validity and relevance of the metrics remains unanswered. [W3]
>
> Given these unresolved issues, I will maintain my current rating.

---

> > ### Author Response · Authors · 2024-12-03
> >
> > Thank you for your interest in our work and your suggestions. Here is our answer to your confusion.
> >
> > **Q1: Unclear Contributions: The contributions of the paper are still not clear. If the plug-and-play feature is a key contribution, it needs to be validated with various existing CTTA methods to demonstrate its generalizability.**
> >
> > A1: It can be clearly stated that plug-and-play is not a contribution of our paper. It only appeared in the rebuttal stage and was merely to illustrate that our method can achieve better results based on CMAE (since the performance of CMAE is very close to that of our method).
> >
> > **Q2: Fairness of Comparison with Baselines: The proposed method assumes stronger conditions compared to some baselines. A thorough discussion of the specific conditions under which C-CoTTA is compared to other methods would be helpful. [W5]**
> >
> > A2: C-CoTTA uses source domain prototypes, so the comparison with RMT and SATA, which also use this method, is completely fair. However, the other methods do not use this, leading to an unfair comparison. However, we firmly believe that in the CTTA scenario, as long as not all data from the source domain is directly used (as this would cause storage pressure and privacy leakage), there is no issue. I wonder how you view this?
> >
> > **Q3: Metric Validity and Relevance: My concern regarding the validity and relevance of the metrics remains unanswered. [W3]**
> >
> > A3: I apologize for previously forgetting to respond to this concern. However, regarding your mention of "scaled up by a factor," we believe this does not exist, because for different methods and different domains, we extract prototypes from the same layer of the same backbone. Therefore, the changes in inter-class and inter-domain distance are not caused by scaling, but rather by the test-time adaptation method, which can reflect class separability. Additionally, regarding your point about "such as between the 'Brightness' and 'Contrast' conditions of ImageNet-C, which have a roughly 20% gap in accuracy despite similar distance metrics," we believe this may be related to other factors of test-time adaptation.

---

### Official Review · Reviewer_MZKq · 2024-11-03

**Soundness:** 3
**Presentation:** 3
**Contribution:** 3
**Rating:** 5
**Confidence:** 4

**Summary:**

Continual Test-Time Adaptation (CTTA) - While most CTTA methods focus on suppressing domain shifts, this paper introduces an approach that aims to guide rather than suppress these shifts. They propose Controllable Continual Test-Time Adaptation (C-CoTTA), designed to maximise inter-class distances while minimising inter-domain distances.

**Strengths:**

The method demonstrates empirically plausible results, though with reservations that I describe in the weaknesses.

Since this area is outside my expertise, I would defer to my fellow reviewers on the following:

- The relevance of the benchmark tasks and datasets used in this work
- The significance of the reported results
- Any potential biases or issues in the experimental setup

**Weaknesses:**

The terms "guide" and "control" are vague. More precise language is needed to clearly convey the conceptual mechanism of C-CoTTA. Specifically, could the authors clarify whether the method is fundamentally maximising inter-class distances and minimising inter-domain distances in the representation space? This objective appears consistent with the aims of many test-time adaptation methods. Detailed technical explanation on how C-CoTTA differs from prior approaches would be beneficial, especially avoiding abstract terms like "suppress," "guide," or "control."

It may not be fair to summarise all baseline methods (Tables 1, 2, and 3) as merely "suppressing the shift." Is it accurate to say that these methods do not involve any "guiding" or "controlling" aspects? Further technical support for this claim would strengthen the paper. Perhaps the authors could include an analysis in §4.5 and Figure 4 for all baseline methods listed in Tables 1, 2, and 3.

The terms "guiding" and "controlling" remain ambiguous, especially given that Equation 7 still seems focused on "suppressing the shift" as well. Clear definitions of these terms within the context of C-CoTTA are recommended.

Since many methods are compared in the benchmark with rather small performance gaps, I wonder about the possibility of test-set overfitting. This is especially important given the low-resource nature of the setup (where information about the test distribution is supposed to be only partially available to the learner). It would be helpful if the authors could describe their efforts to minimise this risk (e.g., by confirming if the hyperparameters and design choices in §C were tuned without reference to the test/evaluation split). This would impact the fairness of the empirical comparisons, given the small performance gaps (15.3 vs 14.7 in Table 1, 30.2 vs 29.9 in Table 2, 59.9 vs 59.4 in Table 3).

nit: please run spell/grammar checker
- "Ativation" --> "Activation"
- "only suppress domain shift is insufficient" --> "only suppressing domain shift is insufficient"
- "explicit control" --> "explicitly control"

**Questions:**

The conceptual contribution of this work remains unclear. As described, it appears to be a rephrasing of the inter-class distance maximisation and inter-domain distance minimisation approach commonly used in CTTA. How does this approach differ meaningfully from existing methods in CTTA?

While the results slightly outperform previous approaches on average, I have concerns about whether hyperparameter tuning was conducted in a way that avoids overfitting to the evaluation benchmark. Please clarify this in the rebuttal.

My current assessment leans toward rejection. I encourage the authors to address these points in their response.

---

> ### Author Response · Authors · 2024-11-15
>
> Thank you for your valuable comments and kind words to our work. Below we address specific questions.
>
> **Q1. How does this approach differ meaningfully from existing methods in CTTA**
>
> A1. In general, our approach is not fundamentally about maximizing inter-class distance and minimizing inter-domain distance in the representation space; in other words, this is not our starting point, but it does achieve these goals. The starting point of our method is to suppress the opposing shifts between categories in the representation space. We discovered through t-SNE feature dimensionality reduction that during the domain adaptation process, opposing shifts occur between categories, leading to blurred classification boundaries, and the situation worsens with continuous transformations of the domain. Therefore, we explicitly control the direction of category shifts to prevent them from occurring in opposition, ensuring a clear classification boundary. (This is what "Controllable" means in our method). Next, to demonstrate that we have achieved this, we plotted t-SNE graphs while comparing the inter-class distances of different methods (because theoretically, if categories do not shift in opposition but rather move away from each other, the inter-class distance should increase; thus, inter-class distance serves as an auxiliary tool to validate our method rather than being the starting point).
>
> **Q2. While the results slightly outperform previous approaches on average, I have concerns about whether hyperparameter tuning was conducted in a way that avoids overfitting to the evaluation benchmark.**
>
> A2. We have a total of three hyperparameters, among which the entropy threshold $E_0$ is directly taken from the parameters of EATA without any adjustment process, while $\lambda_1$ and $\lambda_2$ are derived from adjusting based on 4 different domains other than the 15 domains used in the experiments. For example, the CIFAR-10-C dataset actually has 19 types of corruption (different corruptions represent different domains), and our parameters were adjusted based on 4 corruptions (Speckle Noise, Gaussian Blur, Spatter, Saturate). In the formal experiments, we selected 15 completely different domains (Gaussian Noise, Shot Noise, Impulse Noise, Defocus Blur, Glass Blur, Motion Blur, Zoom Blur, Snow, Frost, Fog, Brightness, Contrast, Elastic Transformation, Pixelate, JPEG).

---

> > ### Comment · Reviewer_MZKq · 2024-11-15
> >
> > Thanks for the quick response. However, I'm not convinced yet. I wish the authors answer the question a bit more directly. Here's the logic:
> >
> > 1. Authors wrote in the abstract:
> > > Existing CTTA methods primarily focus on suppressing domain shifts, which proves inadequate during the unsupervised test
> > phase
> >
> > 2. But then Equation 7 of the paper seems to be doing precisely the suppression of domain shifts that the author criticised.
> >
> > 3. I'm left wondering what difference the proposed method has against the prior work.
> >
> > For Q2, my question is rather about the fairness of comparison - I wonder how much HP tuning was done for the proposed method in comparison to the previous methods. Since the delta in performance is small, I'd expect the degree of HP tuning can easily change the results. I hope authors can defend their position based on numbers, if possible, instead of conceptual, high-level answers!

---

> ### Author Response · Authors · 2024-11-15
>
> Thank you for the quick response. This time I understand your core concerns.
>
> 1. The DSCL loss represented by Equation 7 indeed suppresses domain shifts, reducing the model's sensitivity to domain shifts. However, our focus on "Controllable" mainly revolves around the CSCL loss, which explicitly controls the direction of category shifts, preventing the blurring of classification boundaries caused by opposing category shifts. This loss is the biggest difference from previous work and can be considered the main loss of our paper.
>
> 2. First, our hyperparameter tuning was only performed on the validation set, specifically to find the values of $\lambda_1$ and $\lambda_2$ that yield the best classification performance on the validation set, as shown in Appendix C. Methods like RMT and SATA also performed similar tuning. The reason for the small performance differences may be related to the single order of the domain. Most previous work on CTTA adopted a single order, which may not reasonably assess the performance of various methods. Therefore, we supplemented our study with new experiments (section 4.8). We randomly selected 10 different domain variation orders for testing, and the final results are averaged. It can be seen that in this case, our improvement is significant (1.6%, 3.1%, and 2.4% on CIFAR10-C, CIFAR100-C, and ImageNet-C, respectively). The table is shown in the figure below (which is Table 7 in the paper).
>
>  | Avg. Error |  Source | TENT | CoTTA | SATA | Ours |
>  | ----- | ---- | ---- | ---- | ---- | ---- |
>  | CIFAR10-C   | 43.5     | 20.1   | 16.3      |  16.3     |  14.7 |
>  | CIFAR100-C  | 46.4     | 61.3   | 32.6      |  32.8     |  29.5 |
>  | ImageNet-C  | 83.0     | 61.8  | 57.9      |  64.5     |  55.5 |
>
> If there are any parts of my response that still confuse you, please point them out, and I will reply promptly.

---

> > ### Comment · Reviewer_MZKq · 2024-11-25
> >
> > Thanks for the follow-up. And sorry for my late reply.
> >
> > 1 - Still not sure how meaningful the authors "focus" on CSCL loss is, given that the improvement due to CSCL (as opposed to DSCL) is not strikingly good (Table 4). Is this really a critical ingredient that enables TTA?
> >
> > 2 - Tuning HP on validation set sounds good. But again, the question is more about how the baseline approaches did that and whether the amount of resources put is fair across methods. "Methods like RMT and SATA also performed similar tuning." needs to be substantiated further. Random ordering sounds reasonable, but independent of that it would be great if the authors can add more details about how HP tuning is done for the methods in the benchmark comparison.

---

> ### Author Response · Authors · 2024-11-25
>
> Thank you for your interest in our work and your suggestions. Here is our answer to your confusion
>
> 1. We have always believed that CSCL loss is crucial for the TTA task because it can effectively control the shift in categories, preventing the blurring of classification boundaries caused by opposing shifts between categories, which leads to a decline in performance. As shown in Figure 3 of the article, both CoTTA and SATA methods exhibit blurred classification boundaries. For the CTTA task, continuous domain transformations result in increasingly blurred classification boundaries and accumulating errors, making CSCL even more necessary. We tested the scenario of cycling through 15 domains 10 times, and the experimental results are shown in Table 6 and Figure 5. It can be observed that our method improves classification performance as the domains are repeatedly encountered, ultimately achieving an average error rate that is significantly lower than that of other methods.
>
> 2. After our confirmation, there are no hyperparameters in CoTTA and SATA, and no methods for hyperparameter tuning are provided in RMT and EcoTTA. They only indicate the sensitivity of model performance to hyperparameters on the test set. However, the final reported results all use the best-performing hyperparameters, which is actually unreasonable.
>
> If there are any parts of my response that still confuse you, please point them out, and I will reply promptly.

---

> > ### Author Response · Authors · 2024-12-03
> >
> > Thanks again for your insightful comments and valuable time devoted to our paper. As the author-reviewer discussion period is coming to an end, please let us know if there's any further information we can provide to facilitate the discussion process. We are happy to answer any questions or concerns you may have during the author-reviewer discussion period.

---

### Official Review · Reviewer_3bv1 · 2024-11-04

**Soundness:** 3
**Presentation:** 3
**Contribution:** 2
**Rating:** 6
**Confidence:** 3

**Summary:**

This paper addresses the problem of continual test-time adaptation (TTA) by guiding and controlling shifts by utilizing Concept Activation Vectors (CAV), an interpretability tool for deep learning models.
They control the domain shift by constructing the domain and class shift to control losses.
Experiments show better performance in widely used continual TTA benchmarks.

**Strengths:**

* Utilizing CAV for the problem of continual TTA
* Proposing domain and class shift controlling losses
* Improving performance on the state-of-the-art benchmarks

**Weaknesses:**

* Using pseudo-labels for prototype computation can lead to more error accumulation in real-world
* Motivation for why CAV makes sense for continual TTA and not any deep learning problem, in general, is lacking
* Comparisons with recent approaches such as EcoTTA [1], and BeCoTTA [2] are missing

**Questions:**

* How are the hyperpaprameters lambda_1, lambda_2 tuned? Is any validation corruption used?
* Can you report numbers for the two settings when i. lambda_1 = 0, and ii. lambda_2 = 0 to analyze the contribution of different losses.
* Is the paper not just an application of CAV for continual TTA? Is it non-trivial to apply CAV for settings such as TTA?
* Are the DSCL and CSCL losses not applicable to domain adaptation problems in general? If not, why does it make sense for TTA?

**Minor Comments**
* Line 82-83: Typo "Concept Ativation Vectors" --> "Concept Activation Vectors"
* Algorithm 1 does not define p^t_i
* Some other typos, please proofread

**References**
1. Junha Song, Jungsoo Lee, In So Kweon, and Sungha Choi. Ecotta: Memory-efficient continual test-time adaptation via self-distilled regularization. In Proceedings of the IEEE/CVF Conference on Computer Vision and Pattern Recognition, pages 11920–11929, 2023.
2. Lee, Daeun, Jaehong Yoon, and Sung Ju Hwang. "BECoTTA: Input-dependent Online Blending of Experts for Continual Test-time Adaptation." International Conference on Machine Learning, 2024.

---

> ### Author Response · Authors · 2024-11-15
>
> Thank you for your valuable comments and kind words to our work. Below we address specific questions.
>
> **Q1. How are the hyperpaprameters lambda_1, lambda_2 tuned? Is any validation corruption used?**
>
> A1. $\lambda_1$ and $\lambda_2$ are derived from adjusting based on 4 different domains other than the 15 domains used in the experiments. For example, the CIFAR-10-C dataset actually has 19 types of corruption (different corruptions represent different domains), and our parameters were adjusted based on 4 corruptions (Speckle Noise, Gaussian Blur, Spatter, Saturate). In the formal experiments, we selected 15 completely different domains (Gaussian Noise, Shot Noise, Impulse Noise, Defocus Blur, Glass Blur, Motion Blur, Zoom Blur, Snow, Frost, Fog, Brightness, Contrast, Elastic Transformation, Pixelate, JPEG).
>
> **Q2. Can you report numbers for the two settings when i. lambda_1 = 0, and ii. lambda_2 = 0 to analyze the contribution of different losses.**
>
> A2. This is actually our ablation experiment (Table 4), where CSCL represents the case of $\lambda_1 = 0$ and DSCL represents the case of $\lambda_2 = 0$.
>
> **Q3.  Is the paper not just an application of CAV for continual TTA?**
>
> A3. No, CAV is just a tool we use in our method to represent class shift and domain shift; it plays a supportive role and is not the core of our method. The starting point of our method is to suppress the opposing shifts between categories in the representation space. We discovered through t-SNE feature dimensionality reduction that during the domain adaptation process, opposing shifts occur between categories, leading to blurred classification boundaries, and the situation worsens with continuous transformations of the domain. Therefore, we explicitly control the direction of category shifts to prevent them from occurring in opposition, ensuring a clear classification boundary. (This is what "Controllable" means in our method). In this context, CAV is used to represent the direction of class shift.
>
> **Q4. Is it non-trivial to apply CAV for settings such as TTA? Are the DSCL and CSCL losses not applicable to domain adaptation problems in general? If not, why does it make sense for TTA?**
>
> A4.  First of all, it can be confirmed that the methods in our article, including CAV, DSCL, and CSCL, can potentially be applied to TTA (Test-Time Adaptation) and even more broadly to domain adaptation. However, we did not conduct relevant tests because our initial research focus was on CTTA (Continuous Test-Time Adaptation). Considering that in CTTA, the domain continuously changes, the classification boundary blurriness caused by category shifts is likely to be more severe than in regular TTA. Therefore, our methods may be more suitable for CTTA. Of course, prompted by the reviewers, we also hope to examine the effectiveness of our methods in TTA and domain adaptation. We attempted to apply our methods in the TTA scenario (i.e., recovering the model when the domain changes), as shown in the table below, and it can be seen that our methods still perform very well.
>
> |       | Gauss | Shot | Impul | Defoc | Glass | Motion | Zoom | Snow | Frost | Fog  | Brit | Contr | Elast | Pixel | JPEG | Avg  |
> | ----- | ----- | ---- | ----- | ----- | ----- | ------ | ---- | ---- | ----- | ---- | ---- | ----- | ----- | ----- | ---- | ---- |
> | Tent  | 52.6  | 52.1 | 53.5  | 52.9  | 47.7  | 56.7   | 47.5 | 10.5 | 28.6  | 67.2 | 74.4 | 67.3  | 50.7  | 66.3  | 64.6 | 52.8 |
> | CoTTA | 40.6  | 37.8 | 41.7  | 33.7  | 29.5  | 43.8   | 35.6 | 38.1 | 43.3  | 59.2 | 70.5 | 59.3  | 40.1  | 57.9  | 59.7 | 46.1 |
> | Ours  | 37.1  | 36.1 | 41.2  | 31.7  | 28.4  | 42.5   | 35.1 | 37.2 | 41.2  | 57.2 | 77.2 | 56.4  | 39.2  | 56.8  | 56.4 | 44.9 |
>
>
>
> If there are any parts of my response that still confuse you, please point them out, and I will reply promptly.

---

> > ### Comment · Reviewer_3bv1 · 2024-11-26
> > **Acknowledgement, Comparisons missing**
> >
> > Thanks for the response addressing most of my queries.
> >
> > However, one point that still remains unaddressed or not justified is why comparisons with recent state-of-the-art approaches such as EcoTTA [1] and BeCoTTA [2] are missing.

---

> > > ### Author Response · Authors · 2024-11-27
> > >
> > > Thank you for your interest in our work and your suggestions. Here is our answer to your confusion.
> > >
> > > **Q1: why comparisons with recent state-of-the-art approaches such as EcoTTA [1] and BeCoTTA [2] are missing.**
> > >
> > > A1.  In the rebuttal stage, we added comparisons with the latest SOTA methods (ViDA, CMAE, EcoTTA, Becotta). Additionally, to better compare the performance of each method and eliminate random errors, we used 10 different domain transformation sequences. The results are shown in the table below.
> > >
> > > |         | cifar10-c | cifar100-c | Imagenet-c |
> > > | ------- | --------- | ---------- | ---------- |
> > > | Source  | 43.5      | 46.4       | 83.0       |
> > > | Tent    | 20.1      | 61.3       | 61.8       |
> > > | CoTTA   | 16.3      | 32.6       | 57.9       |
> > > | SATA    | 16.3      | 32.8       | 64.5       |
> > > | RMT     | 16.2      | 32.8       | 63.8       |
> > > | DSS     | 17.5      | 33.4       | 67.5       |
> > > | SWA     | 15.8      | 31.5       | 60.9       |
> > > | PETAL   | 17.1      | 35.6       | 69.5       |
> > > | RoTTA   | 20.1      | 34.9       | 68.5       |
> > > | ViDA    | 15.1      | 30.5       | 57.3       |
> > > | Ecotta  | 16.8      | 33.8       | 62.5       |
> > > | Becotta | 15.8      | 31.6       | 58.9       |
> > > | CMAE    | 14.5      | 29.9       | 56.5       |
> > > | C-CoTTA | 14.7      | 29.5       | 55.5       |
> > >
> > > It can be observed that our method shows improvements compared to other methods. At the same time, we note that the performance of the CMAE method is relatively close to ours. To test the effectiveness of our method, we added our two losses, CSCL and DSCL, based on CMAE (our two losses are plug-and-play). The results are shown in the table below, indicating that our method can further improve CTTA performance.
> > >
> > > |                | cifar10-c | cifar100-c | imagenet-c |
> > > | -------------- | --------- | ---------- | ---------- |
> > > | CMAE           | 14.5      | 29.9       | 56.5       |
> > > | CMAE+CSCL      | 14.1      | 29.1       | 55.5       |
> > > | CMAE+DSCL      | 14.3      | 29.4       | 56.1       |
> > > | CMAE+CSCL+DSCL | 13.9      | 28.8       | 55.2       |
> > >
> > >
> > >
> > > If there are any parts of my response that still confuse you, please point them out, and I will reply promptly.

---

### Author Response · Authors · 2024-12-01

Thanks again for your insightful comments and valuable time devoted to our paper. As the author-reviewer discussion period is coming to an end, please let us know if there's any further information we can provide to facilitate the discussion process. We are happy to answer any questions or concerns you may have during the author-reviewer discussion period.

---

### Note · Authors · 2024-12-24

I have read and agree with the venue's withdrawal policy on behalf of myself and my co-authors.